# Use of PCR Cycle Threshold and Clinical Interventions to Aid in the Management of Pediatric *Clostridioides difficile* Patients

**DOI:** 10.3390/microorganisms12061181

**Published:** 2024-06-11

**Authors:** Mohammed Suleiman, Patrick Tang, Omar Imam, Princess Morales, Diyna Altrmanini, Kelli L. Barr, Jill C. Roberts, Andrés Pérez-López

**Affiliations:** 1Department of Pathology, Sidra Medicine, Doha P.O. Box 26999, Qatar; ptang@sidra.org (P.T.); pmorales@sidra.org (P.M.); daltrmanini@sidra.org (D.A.); aperezlopez@sidra.org (A.P.-L.); 2Department of Pathology and Laboratory Medicine, Weill Cornell Medicine in Qatar, Doha P.O. Box 24144, Qatar; 3Department of Infectious Diseases, Sidra Medicine, Doha P.O. Box 26999, Qatar; oimam@sidra.org; 4Center for Global Health and Infectious Disease Research, University of South Florida, Tampa, FL 33612, USA; barrk@usf.edu (K.L.B.); jcrobert@usf.edu (J.C.R.)

**Keywords:** *Clostridioides difficile*, pediatric, children, cycle threshold, intervention

## Abstract

Better diagnostic tools are needed to improve the diagnosis of *Clostridioides difficile* infections (CDI) and reduce the overtreatment of colonized children. In this study, we evaluated two polymerase chain reaction (PCR) assays (Cepheid GeneXpert *C. difficile* and the Gastroenteritis PCR Panel by QIAstat-Dx) as a standalone method in combination with the PCR cycle threshold (Ct) value in positive samples to predict the presence of free toxins. We also evaluated the clinical impact of reporting toxin production results and provided comments alongside the PCR results in our pediatric population. PCR-positive stool samples from pediatric patients (aged 2 to 18 years old) were included in our study and tested for the presence of toxins A and B using the *C. difficile* Quik Chek Complete kit. For the clinical intervention, the CDI treatment rates 6 months pre- and post-intervention were compared. The use of PCR Ct value showed excellent sensitivity (100%) at a Ct value cutoff of 26.1 and 27.2 using the Cepheid GeneXpert *C. difficile* and the Gastroenteritis PCR Panel by QIAstat-Dx, respectively, while the toxin test showed inferior sensitivity of 64% in the PCR-positive samples. In addition, CDI treatment rates were decreased by 23% post-intervention. The results of our study suggest that nucleic acid amplification test (NAAT) assays supplemented by the use of PCR Ct value for positive samples can be used as standalone tests to differentiate CDI from colonization. Furthermore, the reporting of toxin production along with the PCR results can help reduce the unnecessary treatment of colonized children.

## 1. Introduction

*Clostridioides difficile* is a major public health concern as it is one of the main causes of diarrhea in hospitalized patients and an increasing cause of community-acquired diarrhea [1,2]. *C. difficile* infections (CDIs) have been associated with longer hospital stays, increased healthcare-associated costs, and increased risk of death in children [2,3,4,5]. Prompt and accurate diagnosis of CDI is essential in pediatric hospitalized patients to ensure that proper treatment and infection control measures are in place [1,2]. Although many diagnostic methods and algorithms are used across laboratories, including glutamate dehydrogenase (GDH), toxin enzyme immunoassays (EIAs) and nucleic acid amplification tests (NAATs), there is no single method or algorithm that is universally accepted [2,6]. The diagnosis of CDI is further complicated in the pediatric population by the high prevalence of asymptomatic carriage of toxigenic and non-toxigenic strains which makes it difficult to distinguish colonization from infection in children [3]. There are many FDA-approved NAATs available, either as standalone assays or within multiplex syndromic PCR panels, to detect the genes encoding *C. difficile* toxins (*tcdA* and *tcdB*) [2,5]. These molecular assays are more sensitive for the detection of *C. difficile* than traditional GDH and toxin EIAs. However, they only detect the presence of genes encoding the *C. difficile* toxins, they lack enough specificity to rule in CDI, and are unable to differentiate between symptomatic infection and colonization [2,5]. There is a very limited number of *C. difficile* studies involving pediatric patients [3,7]. Furthermore, many of the studies are limited by the lack or inappropriate use of reference diagnostic methods, lack of knowledge about the prevalence of CDI in the studied population, and lack of clinical assessment and/or clinical intervention as part of the study [3].

Better diagnostic tools are needed to improve the management of pediatric patients suspected of having CDI and to help in distinguishing between colonization and true infection. There is no question about the superior diagnostic accuracy of the NAAT assay in detecting the genes encoding for *C. difficile* toxins (*tcdA* and *tcdB*), but there are still many questions raised regarding the clinical significance of NAAT testing. Using the clinical review of pediatric patients as a reference method, we investigated the option of using PCR (Cepheid GeneXpert *C. difficile* or the Gastroenteritis PCR Panel by QIAstat-Dx) as a standalone method with the assessment of Ct value in positive samples to predict the presence of free toxins to guide the diagnosis of CDI in pediatric patients. We also evaluated the clinical impact of reporting toxin production results alongside the PCR results in our pediatric population, especially in reducing the overtreatment of colonized children.

## 2. Materials and Methods

### 2.1. Ethics

This study was approved by the Internal Review Boards at Sidra Medicine (2042490) and the University of South Florida (STUDY005722).

### 2.2. Use of PCR Ct Value

#### Specimen Collection and Study Design

PCR-positive, fresh stool samples, soft or liquid, collected prospectively from pediatric patients (aged 2 to 18 years old) submitted to our microbiology laboratory for routine gastrointestinal PCR panel or *C. difficile* testing at Sidra Medicine between June and December 2023 were included in this study. Sidra Medicine is a single-center pediatric tertiary care hospital in the State of Qatar, which is the main pediatric subspecialty referral center in the country. All PCR-positive pediatric samples collected from all inpatient units, emergency services, and outpatient units were included in our study. Samples were collected in sterile containers and testing was performed immediately upon receipt in the microbiology laboratory except for samples received overnight which were stored at 2–8 °C and tested the following day. According to our standard laboratory rejection criteria, formed stools, repeat samples within seven days from the same patient, and repeat PCR positives within 30 days from the same patient were excluded.

### 2.3. Clinical Review

Clinical data were collected including patient age, patient sex, patient location, previous history of CDI, previous or current antibiotics used, white blood cell count, and clinical symptoms such as fever and diarrhea. A clinical review of the patient chart was performed to determine if the patient had a true CDI. In the clinical review, a patient was considered as true positive if they had diarrhea (>3 loose stools) without an alternative diagnosis, and at least one of the following risk factors: antibiotic exposure in the last 6 weeks, immunosuppressive therapy, oncology condition, solid organ transplantation, and inflammatory bowel disease (IBD).

### 2.4. C. difficile Laboratory Diagnostic Tests

All samples were tested prospectively for the presence of toxins A and B using the *C. difficile* Quik Chek Complete kit (CDQ; Techlab, Blacksburg, VA, USA), detection of toxin B gene using the Xpert *C. difficile* PCR test (XCD; Cepheid, Sunnyvale, CA, USA), and detection of toxin A and B genes using the QIAstat-Dx Gastrointestinal Panel 2 PCR test (QGP; QIAGEN, Hilden, Germany). All assays were performed, and results were interpreted according to each manufacturer’s instructions.

### 2.5. Statistical Analysis

We used clinical review as a reference method. The PCR Ct cutoff value was determined using the Youden maximum index while fixing the sensitivity to ≥99%. The sensitivity, specificity, positive predictive value (PPV), and negative predictive value (NPV) were calculated with 95% confidence intervals for the XCD with a Ct cutoff of 26.1, QGP with a Ct cutoff of 27.2, and the toxin result of the CDQ kit.

### 2.6. Clinical Intervention

The laboratory intervention study used samples submitted for *C. difficile* PCR testing by XCD with the same sample inclusion and exclusion criteria mentioned above. Samples tested by QGP alone were excluded from this part of the study, as this method was not available in the pre-intervention period. Retrospective data were collected for PCR-positive samples that were tested using XCD between 1 January 2023 to 24 June 2023 (pre-intervention period). On 25 June 2023 and until 16 December 2023 (post-intervention period), we started reporting the toxin production results based on the CDQ kit in the electronic medical record (EMR) along with the XCD PCR results. The toxin production results were reported along with a comment encouraging physicians to correlate the results with the clinical symptoms before starting therapy if the patient was PCR-positive but negative for toxin production. In addition, a follow-up infectious diseases clinical consultation was performed on all PCR-positive but toxin-production-negative results. Clinical data collected (pre- and post-intervention) included patient age, patient sex, patient location, and CDI treatment used (oral vancomycin or metronidazole). We compared the rate of antibiotic prescribing for CDI treatment pre- and post-intervention periods regardless of toxin production status. Chi square analysis was performed to determine statistical significance.

## 3. Results

### 3.1. Use of PCR Ct Value

A total of 49 PCR-positive stool samples meeting the inclusion criteria of our study were tested using all three testing methods. The median age of patients with PCR-positive samples was 8 years old. In total, 30 (61.2%) of the patients were male and 19 (38.8%) were female; 20 (40.8%) of the samples originated from the hospital units, while 8 (16.3%) were from the emergency department and 21 (42.9%) were from outpatient clinics. In the clinical review, 27 (55.1%) patients had previous antibiotic exposure, 3 (6.1%) were cancer patients, 9 (18.4%) were IBD patients, and 3 (6.1%) patients had solid organ transplant.

The XCD PCR did not detect any NAP1 strains. Among the PCR-positive samples, the median Ct value for samples tested using XCD and QGP was 30.1 and 28.7, respectively. Figure 1 and Figure 2 show the distribution of the PCR Ct values obtained by samples tested by XCD and QGP methods. Five samples were not detected by QGP and two samples were not detected by XCD. Figure 3 shows the correlation between the Ct values obtained using both methods. The sensitivity, specificity, PPV, and NPV for the XCD at a Ct cutoff of 26.1, QGP at a Ct cutoff of 27.2, and the toxin method part of the CDQ kit is detailed in Table 1. The toxin method yielded excellent specificity and PPV (both 100%) but showed inferior sensitivity (64%) and NPV (88%) (Table 1). Both PCR methods with the proposed Ct cutoffs yielded excellent sensitivity and NPV (both 100%) with lower specificity (94% XCD, 89% QGP) and lower PPV (88% XCD, 78% QGP) (Table 1).

### 3.2. Clinical Intervention

Pre-intervention period: In total, 56 patients with XCD PCR-positive results from 1 January 2023 to 24 June 2023 were reviewed. After reporting the positive XCD PCR result, 38 patients (68%) were treated for CDI while 18 patients (32%) were not treated. In total, 36 patients were treated with oral vancomycin and two patients were treated using oral metronidazole

Post-intervention period: In total, 40 patients with XCD PCR-positive results from 25 June 2023 to 16 December 2023 were reviewed. Following the positive XCD PCR result, 18 patients (45%) were treated for CDI while 22 patients (55%) were not treated. In total, 16 patients were treated with oral vancomycin and two patients were treated with oral metronidazole. Overall, there was a reduction of 23% (*p*-value = 0.025) in antibiotic treatment between the pre- and post-intervention periods.

## 4. Discussion

CDI diagnosis remains a significant challenge in the pediatric population as it is difficult to distinguish colonization from infection in young children. The Infectious Diseases Society of America (IDSA) and the Society for Healthcare Epidemiology of America (SHEA) recommend using stool toxin test as part of a multistep algorithm (e.g., glutamate dehydrogenase (GDH) in combination with toxin detection method) or GDH plus toxin, arbitrated by nucleic acid amplification test (NAAT) [3,8]. IDSA-SHEA also recommends using NAATs alone as a first-step method if stool samples from patients with suspected CDI have been previously screened (patients with clinical diarrhea and a history of previous antibiotic exposure who have not received any laxatives in the past 48 h) [3,8]. Although NAAT tests have high analytical sensitivity, they lack clinical specificity to differentiate between colonization and infection. In addition, recent evidence supports the use of *C. difficile*-free toxin detection to predict CDI outcomes and help guide treatment decisions [9,10,11].

Our study results show that we can improve the clinical accuracy of CDI testing by using the NAAT method alone followed by the assessment of Ct value to predict the production of free toxins and to differentiate between colonization and infection in PCR-positive samples. Using a stand-alone PCR method is a very popular option and has been widely adopted in many clinical laboratories in the United States [12]. Adding the use of PCR Ct value for these laboratories can improve the sensitivity and specificity of their results without significantly changing the workflow or incurring additional costs. Our results show that the PCR Ct value can accurately predict all true positive CDIs (100% sensitivity for both NAAT methods). If we included all negative PCR samples (n = 311) tested during the same period, the specificity of the NAAT tests would increase to 99% using XCD or QGP methods. One previous study, which evaluated the performance of the *C. difficile* PCR Ct value for predicting free toxin status (in a population that included 63 pediatric patients), had very similar results but it did not evaluate the clinical performance of the Ct toxin cutoff [13]. This study showed the PCR Ct value method to be highly sensitive for CDI at a PCR Ct value cutoff of 26.4 with sensitivity, specificity, PPV, and NPV of 96.0%, 65.9% (improved to 78% by including CTA as a reference method), 57.4%, and 97.1%, respectively [13].

In addition, the results of our study showed that using a stand-alone PCR method supplemented by reporting PCR Ct value to predict the presence of free toxin production is more sensitive than using traditional toxin EIA kits (toxin EID sensitivity is 64% vs. 100% for NAAT) (Table 1). These data support the use of NAAT alone instead of using a multi-step algorithm combining NAAT with toxin EIAs. Also, these results have the potential to reduce antibiotic treatment for CDI in up to 67% of PCR-positive (with a Ct value > 26.1) patients in our population.

In contrast to the adult population, there are a limited number of *C. difficile* studies involving pediatric patients evaluating the use of PCR Ct value and its association with clinical outcomes. A few studies in the adult population found no association between lower PCR Ct values and severe CDI or poor clinical outcomes [14,15,16]. On the other hand, many more studies showed an association between lower PCR Ct values and severe CDI and poor clinical outcomes including mortality, symptoms, and length of stay [17,18,19,20,21].

During our study, we did not report the PCR Ct value in our EMR system as this approach was not yet validated. Instead, we reported the results of the toxin EIA kit which was already validated and in use for routine patient testing. The results of clinical intervention evaluated as part of our study suggested that reporting the toxin production results with the PCR results reduced the percentage of PCR-positive pediatric patients treated for CDI. The number of PCR-positive tests that resulted in CDI treatment significantly decreased by 23% (*p*-value = 0.025) after the implementation of this intervention. It can also be speculated that if the clinical intervention had been performed based on the Ct value instead of toxin detection, the number of patients treated would have been even lower. The result of our intervention is similar to that of a previous study which showed that adding the *C. difficile* PCR Ct value result to PCR reporting reduced the proportion of PCR-positive children who are treated for CDI by 22% post-intervention [9]. The study was performed on 292 pediatric patients (1–18 years old) and used the Cepheid Xpert *C. difficile* assay with a PCR Ct cutoff of ≤27.5 to determine toxin positivity [9]. Another study in the adult population showed that dual reporting of *C. difficile* PCR and the predicted toxin result based on PCR Ct value reduced the treatment of toxin-negative patients without increases in adverse outcomes [22]. By inferring toxin production from the PCR Ct value, 51.5% of toxin-negative patients were treated compared with 98.2% of toxin-positive patients being treated [22].

Our study excluded pediatric patients younger than 2 years old due to the high prevalence of asymptomatic carriage of toxigenic *C. difficile* among this age group, which is in line with the IDSA and SHEA recommendations [3]. This is also supported by two recent studies that showed that the use of the PCR Ct value in this age group cannot differentiate CDI from colonization [23,24].

A limitation of our study is the low number of positive samples as we have a low prevalence of *C. difficile* in our pediatric population (5%). Another limitation is that we did not evaluate pre-analytical factors that could affect the PCR Ct value such as stool quality, stool volume, and immune status of patients. A previous study evaluated several of these preanalytical factors including stool quality, patient’s age, immune status, and strain type and did not find significant change in the median PCR Ct values, suggesting that the use of PCR Ct value cutoff to help in the diagnosis of CDI could be less prone to preanalytical factors as DNA might be more stable than toxin in samples [13].

## 5. Conclusions

The results of our study suggest that NAAT assays can be used as standalone tests followed by the assessment of PCR Ct value for positive samples to diagnose CDI more accurately and differentiate colonization from infection in children. As results may vary based on the method and the *C. difficile* prevalence in each patient population, laboratories considering this approach will need to validate the use of PCR Ct values for CDI diagnosis in their own setting. In addition, the reporting of the toxin production alongside the PCR results reduced the overtreatment of colonized children.

## Figures and Tables

**Figure 1 microorganisms-12-01181-f001:**
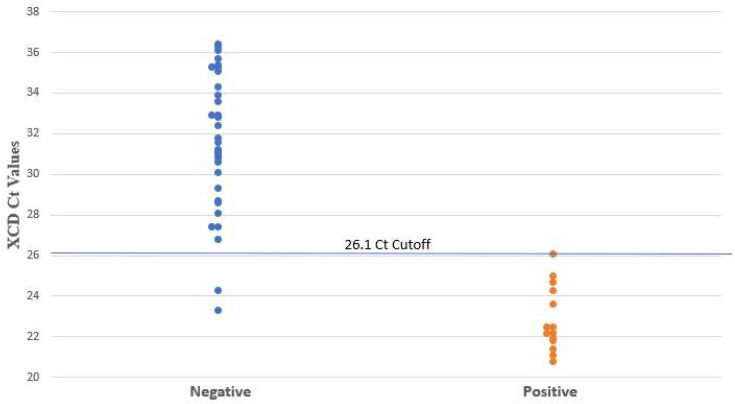
*C. difficile* Ct values determined by XCD. Negative and positive results are based on clinical review.

**Figure 2 microorganisms-12-01181-f002:**
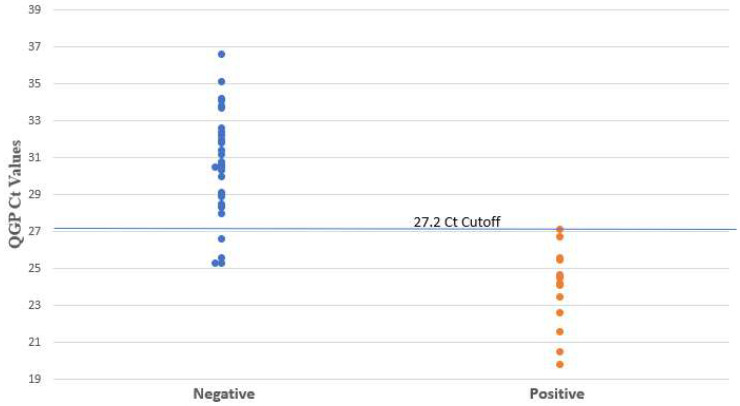
*C. difficile* Ct values determined by QGP. Negative and positive results are based on clinical review.

**Figure 3 microorganisms-12-01181-f003:**
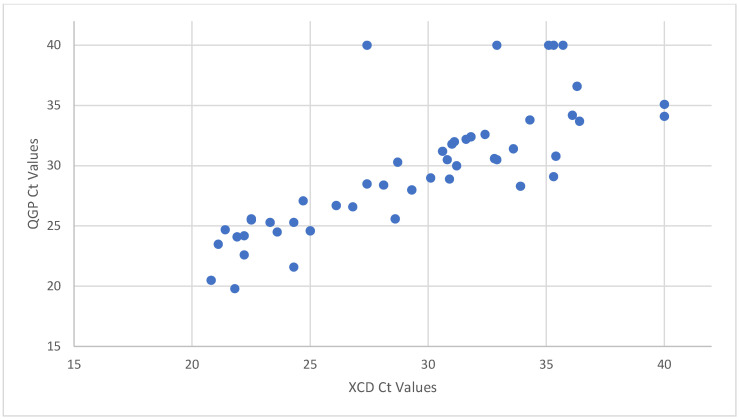
Correlation of PCR Ct values between QGP and XCD. Samples not detected by PCR were assigned a Ct value of 40.

**Table 1 microorganisms-12-01181-t001:** Performance characteristics of toxin A/B EIA and PCR Ct value for XCD and QGP with sensitivity fixed ≥99%.

Method	Ct Cutoff	Reference	TP	TN	FP	FN	Sensitivity (%)	Specificity (%)	PPV (%)	NPV (%)
XCD	26.1	Clinical review	14	33	2	0	100 (78–100)	94 (81–98)	88 (64–97)	100 (90–100)
QGP	27.2		14	31	4	0	100 (78–100)	89 (74–95)	78 (54–91)	100 (89–100)
Toxin A/B EIA	N/A		9	35	0	5	64 (39–84)	100 (90–100)	100 (70–100)	88 (74–96)

Ct, cycle threshold; EIA, enzyme immunoassays; FN, false negative; FP, false positive; N/A, not applicable; NPV, negative predictive value; PPV, positive predictive value; QGP, QIAstat-Dx Gastrointestinal Panel 2 PCR test; TN, true negative; TP, true positive; XCD, Xpert *C. difficile* PCR test.

## Data Availability

The original contributions presented in the study are included in the article, further inquiries can be directed to the corresponding author.

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
