# Peer review of "Use of PCR Cycle Threshold and Clinical Interventions to Aid in the Management of Pediatric Clostridioides difficile Patients"

_microorganisms, 2024, doi:10.3390/microorganisms12061181_

Round 1
Reviewer 1 Report
Comments and Suggestions for Authors
This is a well-written manuscript confirming a previously described observation - that a lower cycle threshold (Ct) of a C difficile tcdB assay predicts the presence of toxin a/b in a stool sample, and presumably is more clinically relevant/necessitating treatment.. This finding is shown for 2 different PCR platforms. Further, it is shown that reporting of the presence of toxin by EIA in addition to a positive PCR result can reduce treatment of C difficile. Finally, using a "clinical review" as the reference gold standard, it was shown that a "lower-than-cutoff Ct" PCR result is more predictive of "true" C difficile disease than the toxin assay alone. This last point is interesting; "true" C difficile was defined by a "clinical review" criterion which required 1. > 3 loose stools/d, 2, no alternate diagnosis for diarrhea (which I assume include use of laxatives), and 3.1 risk factor -prior antibiotics, immunosuppression/oncology, or IBD). IN other words, the toxin assay was missing cases that were PCR-positive and fulfilled the above criteria (and in addition appeared to have low Ct values in the PCR assays). These clinical criteria seem reasonable, though may miss the cases of community-dwelling,, non-antibiotic associated C diff reported in some parts of the US (Chitnis AS, et al Epidemiology of community-associated Clostridium difficile infection, 2009 through 2011. JAMA Intern Med. 2013 Jul 22;173(14):1359-67. doi: 10.1001/jamainternmed.2013.7056. PMID: 23780507..) This ties in to line 32 which states C diff is a "common cause of community-acquired diarrhea" -a claim which is probably overstated as the relative contribution of CDI to all community-acquired diarrhea is quite small.
Other comments:
1.It is not clear from the text how these clinical review cases, numbering 49 compare to the "40 patients with XCD PCR positive results" in the post-intervention period cited in line 145.
2, It would be interesting to note the results of the clinical reviews -eg how many patients had prior antibiotics, immunosuppression ,cancer or IBD. How many were from outpatients or emergency rooms?
3. Minor point-line 39-eliminate the word "but"
4. Fig 1 and 2 need to explain the meaning of negative and positive on the x axis - I assume those refer to the toxin assays?
5.The CT cutoff values need to be standardized - the figures indicate 26.1 and 27.2 but the Table indicates 26.2 and 27.1
6.Table 1 legend- what does sensitivity fixed > 99% mean here
7.What was the correspondence/correlation between the 2 PCR assays in terms of CT values.
Reviewer 2 Report
Comments and Suggestions for Authors
I have read with interest the manuscript submitted by Suleiman et al.
I have some comments to be addressed in order to improve the quality of the manuscript:
- abstract - all abbreviations used should be described at first use
- I recommend editing the citation of the articles using [] instead of ()
- rows 39-42 - I recommend expanding this topic and providing further information on the differential diagnosis between infection and colonization in children
Results - any information about the clinical manifestations of the patients? Risk factors for CDI infection?
- I recommend adding some more robust statistical methods
Discussions - I recommend adding further studies in order to compare the results
The Conclusions should be a different section and should depict the findings and recommendations of the study.
- the reference list is scarce and not edited according to the mdpi pattern
Comments on the Quality of English LanguageMultiple typos/punctuation errors were identified.
Round 2
Reviewer 2 Report
Comments and Suggestions for Authors
I appreciate the author's efforts in addressing my comments.
The quality of the manuscript has improved and may be published in its current form, in my opinion.
Best regards,